# How Superiors Support Employees to Manage Emotional Demands: A Qualitative Study

**DOI:** 10.3390/ijerph22050670

**Published:** 2025-04-24

**Authors:** Lars Peter Andersen, Jesper Pihl-Thingvad, Dorte Raaby Andersen

**Affiliations:** 1Danish Ramazzini Centre, Department of Occupational Medicine, Gødstrup Hospital, 7400 Herning, Denmark; doteader@rm.dk; 2Department of Occupational and Environmental Medicine, Odense University Hospital, 5000 Odense, Denmark; jesper.pihl-thingvad@rsyd.dk

**Keywords:** emotional demands, supervisors, managing emotional demands, burnout

## Abstract

Previous research has found that emotional demands in the workplace can be taxing and contribute to an increased risk of mental health challenges, including burnout and depression. This study examines how supervisory support can assist employees in managing these demands. Against this background, we investigated the ways in which supervisors facilitate employees’ ability to manage emotional demands while fostering trust in the workplace. Drawing on interviews with supervisors and 32 workgroups from 14 different workplaces, we identified both formal and informal practices that support employees. Supervisor-supported practices include the opportunity for supervision; discussions of emotionally demanding patients, citizens, or students; prompt feedback; “venting”; rotating tasks; and discussing strategies for managing high emotional demands. The findings suggest that supervisors and employees largely align their descriptions of the practice, indicating a shared understanding of supportive practices in the workplace. However, some supervisors were unsure whether to take a proactive or reactive approach to supporting their employees. Additionally, some structural constraints were identified, particularly in the form of budget cuts. Supervisors emphasise the significance of trust-building through accessibility, framing mistakes as learning opportunities, and demonstrating employee confidence. This dual approach, which combines practical support with trust-building, underscores the critical role of supervisors in promoting well-being and engagement in emotionally demanding work environments. While there is a risk that supervisors may exaggerate their efforts toward researchers, employee feedback corroborates their claims. Based on these findings, we recommend that organisations operating in emotionally demanding environments allocate sufficient resources to supervisors, enabling them to implement these practices effectively and foster both emotional support and trust in the workplace.

## 1. Introduction

### 1.1. Impact of Emotional Demands

Professions such as nursing, medicine, social work, teaching, and various service-oriented sectors carry significant responsibilities concerning the health, well-being, and overall support of patients, clients, students, and consumers. Employees in these fields are often tasked with engaging individuals who experience traumatic or stressful life events, which exposes them to significant emotional demands [1]. A defining characteristic of these professions is the necessity for sustained social interaction, which requires employees not only to engage with those they serve in a professional capacity, but also to manage the emotions expressed by individuals in distress [2]. This interaction creates emotional demands, especially when engaging with individuals facing severe illness, accidents, grief, crises, or other social adversities, which collectively contribute to the emotional burden experienced by professionals in these roles [3,4]. Several studies have documented that emotional demands can be exhausting and increase the risk of mental health issues, such as burnout [5,6], depression [4,7], and long-term sick leave [8,9]. Furthermore, one study found that workers with high emotional demands were more likely to experience suicidal ideation [10]. The negative impact of emotional demands on employee well-being [11] may also adversely affect work engagement, including vitality [12]. A review conducted by Johnson et al., 2017, found that employees working in mental healthcare facilities experienced lower levels of well-being than those in other healthcare settings [13].

Despite the known impact of emotional demands, limited research has focused on how supervisory support specifically mitigates these effects across various workplace contexts. Therefore, this study aims to explore the practices through which supervisory support can help employees manage emotional demands.

### 1.2. Understanding Emotional Labour

Emotional labour, the process by which employees manage their emotions to fulfil job requirements, is a critical factor in understanding how professionals cope with the emotional demands of their work. Managing emotions is required for many employees in these professions, which is called emotional labour [14]. This requirement arises from the rigorous demands associated with these professions, wherein employees are required to uphold professionalism and empathy and deliver high-quality services. Hochschild’s (Hochschild, 2015) theory of emotional labour comprises two essential components: “feeling rules” and “display rules”. Feeling rules refer to the norms and expectations that govern the emotions employees anticipate. Conversely, display rules delineate how emotions should be externally manifested, regardless of the individual’s authentic emotional condition [14]. Emotional labour involves the utilisation of two distinct emotion regulation strategies: surface acting and deep acting [14]. Surface acting involves feigning or pretending to experience a specific emotion, whereas deep acting entails genuinely experiencing a desired emotion [15]. Research on emotional labour has consistently shown that surface acting negatively impacts job-related outcomes, such as burnout, job satisfaction, and job performance, whereas deep acting often has a positive effect [16,17,18].

### 1.3. The Role of Supervisor Support

While emotional labour strategies, such as surface and deep acting, help employees navigate feelings and display rules, the extent to which these strategies impact occupational stress and well-being is significantly influenced by supervisory support [16,17,18]. Given the serious health consequences that high emotional demands can have on employees, daily supervisory support plays an important role in managing emotional demands and ensuring the well-being of employees. In jobs with high emotional demands, a study found that supervisors can offer assistance with complex tasks, help employees develop clinical judgement, offer regular and high-quality clinical supervision, and offer emotional support during difficult moments [19]. Studies have found that perceived supervisory support positively affects emotional labour [16,17,18]. For instance, one study found that receiving support from supervisors enhances the benefits of deep acting and reduces the drawbacks of surface acting on job satisfaction and burnout [18]. Similarly, a study conducted on Korean Airline employees found that managerial support moderated the relationship between deep acting and job performance [17]. Finally, one study found that supervisor support plays a crucial role in mitigating the negative effects of surface acting on work engagement, well-being, and turnover intentions [20]. Although these studies highlight several important practices that leaders can implement to help employees manage high emotional demands, they also have several limitations. These studies are often quantitative in design. The predominance of quantitative studies in this field limits our understanding of the dynamics and how supervisors can support employees in managing the emotional demands of their work and capturing these complexities. The above-mentioned studies show that supportive leadership fosters a positive work environment and influences whether employees engage in surface or deep acting in response to the emotional demands of their roles. These studies highlight the pivotal role of supportive leadership in workplaces with high emotional demands. While previous studies have documented the negative effects of emotional demands and the positive effects of perceived supervisor support, how supervisors’ support can mitigate these effects in diverse workplace settings remains underexplored. Existing studies fail to provide a detailed understanding of how supervisory behaviours and practices specifically address the complex emotional demands of high-intensity workplaces, leaving a gap in the understanding of how these dynamics unfold across different workplace contexts. This gap can make it challenging to establish consistent guidelines or actionable steps, ultimately complicating efforts to translate interventions aimed at improving supervisor support within organisational settings.

Despite these challenges, qualitative studies have highlighted the importance of such efforts. For instance, a comprehensive review of 13 qualitative studies underscores the necessity and significance of implementing support and supervision mechanisms to assist staff in managing the diverse emotional challenges associated with their professional responsibilities [1]. Other qualitative studies have investigated which factors employees perceive as being associated with health-promoting leadership and supervisory strategies for managing occupational stress and returning to work. For instance, interviews were conducted with 39 employees who had been absent from work for at least three weeks. The findings indicated that supervisors exhibiting empathy and effective communication skills engaged in collaborative efforts to determine suitable work modifications and offered ongoing support. This approach contributed to the employees feeling both confident and valued in their workplace [21]. Furthermore, a longitudinal qualitative study found that employees perceive that supervisors play a key role in enabling workers to perform valuable work [22]. A study among 12 nurses based in Norway found that supervisors need to be attentive to employees, as this is regarded as a prerequisite for seeing, listening, showing care, and providing constructive feedback [23]. Another qualitative study found that employees highly valued leaders who demonstrated a personal commitment and dedication to their work, maintained availability, provided support to the team, and adopted effective management approaches, particularly in handling complex situations [24]. These qualitative studies highlight the crucial role of supervisors in promoting employee well-being and managing occupational stress. Employees value supervisors who actively engage in work adjustments, provide ongoing support, provide genuine care, and are committed and available. However, a limitation of these studies is that most have examined strategies for managing emotional demands primarily from the employee’s perspective. Without understanding supervisors’ perspectives, it becomes difficult to provide actionable guidelines on how to effectively manage their responsibilities in creating supportive environments. Nonetheless, a few studies have addressed this gap by incorporating supervisors’ perspectives. For instance, a study by Rollins et al., 2021, involved interviews with 40 mental health clinicians and managers to explore their perceptions of the organisational conditions influencing burnout and work engagement [25]. The authors identified that fostering a workplace culture centred on person-focused care, rather than solely productivity, while implementing effective strategies to navigate bureaucratic challenges and supporting employee growth and self-care were key factors that could mitigate burnout and enhance engagement. Another study, based on semi-structured interviews with 30 Allied Health Managers, found that an empathetic leadership style—one that seeks to understand and support staff—could improve employee morale within public health organisations and reduce the risk of burnout [26]. Additionally, a study based on semi-structured interviews was conducted with 29 supervisors from 15 organisations in Australia, focusing on the strategies used by supervisors to manage employee stress. The study identified six overarching themes: four that reflect a risk assessment process model (i.e., problem identification, immediate problem execution, coping assistance, and follow-up and evaluation) and two that represent supervisory leadership behaviours that promote prevention and foster an organisational culture supportive of health [27]. Finally, a study investigated and compared stress and resilience factors in the nursing profession from the perspective of registered nurses and their supervisors. The study found that openness, trust, conflict management, and dealing with mistakes were described as requiring development in order to manage stress [28]. While these studies offer valuable insights into supervisors’ perspectives on managing stress, mitigating burnout, and fostering work engagement, a gap remains in understanding how supervisors believe they can effectively address the negative impacts of emotional demands across different workplace contexts. This study seeks to address this gap.

Relying solely on one viewpoint (employees or supervisors) has some limitations. The existing literature points to notable discrepancies between supervisors’ and employees’ perceptions of support, with supervisors often overestimating the support they believe they provide relative to the employees’ perceptions. A recent meta-analysis of aligned perceptions indicated that supervisors’ perceptions of their leadership are predominantly more positive than employees’ perceptions [29]. Therefore, this study explores the perspectives of both supervisors and employees. By examining both viewpoints, our research seeks to uncover a nuanced understanding of supervisory support and trust, offering actionable insights that may be applicable across diverse workplace settings.

### 1.4. Trust Is Essential for Fostering Effective Supervisory Support

To facilitate the receipt of supervisory support, feedback, and recognition, supervisors must establish an environment in which employees perceive them as trustworthy and secure. Therefore, it is essential to create an environment in which employees feel comfortable expressing their emotions, sharing ideas, and taking risks, without fear of negative consequences. One study examined how employees responded to emotionally demanding situations and assessed how their needs for support evolved over time. The study’s results highlight the importance of building a trusting relationship and how supervisors adapted their responses [19]. Additionally, in workplaces with high emotional demands, a previous study found that the expression of emotions serves important functions for social worker team members, as it fulfils a need that cannot easily be met by someone outside the team [30]. A prerequisite for being able to share something as sensitive as feelings, thoughts, and concerns is a sense of trust that one’s expression of emotions will not be ridiculed or ignored. To benefit from expressing emotions, supervisors must limit criticism, anger, and confrontational behaviour.

Therefore, trust plays a crucial role in the relationship between supervisory support and employee benefits. Defined as a psychological state comprising a willingness to accept vulnerability based on positive expectations of the intentions or behaviour of another [31], trust in a supervisor facilitates the positive psychological conditions of meaningfulness, safety, and availability [32]. This highlights the importance of fostering positive supervisor–employee relationships, building trust, and creating a supportive work environment to maximise employee benefits and organisational outcomes.

### 1.5. The Aim of the Study

This study aims to gain a deeper understanding of how supervisors support employees in managing the emotional demands of their work. Using a qualitative approach, we seek to identify the specific actions, behaviours, and practices supervisors employ to provide support, offering detailed insights and actionable strategies to enhance workplace well-being. Given the existing gaps in understanding supervisory support across diverse professions and the need for qualitative insights, our research questions were designed to explore how supervisors can effectively assist employees in managing emotional demands, fostering trust, and developing practices to improve mental health and reduce burnout in high-intensity work environments.

More explicitly, this study examined the following questions:How do supervisors support employees in managing the emotional demands of their work? What specific actions, practices, and behaviours do supervisors use to help employees manage their emotional demands?How do supervisors build and maintain trust to enhance the effectiveness of their support in helping employees manage emotional demands?

## 2. Methods

A qualitative approach was chosen to capture the depth and complexity of supervisory support strategies, allowing for the exploration of the nuanced perceptions and experiences of both supervisors and employees.

### 2.1. Data Collection

The workplace selection was carried out by identifying industries in Denmark with high emotional demands. This information was reported in the National Workplace Environment Monitoring Among Employees [33], which is conducted biennially and invites approximately 50,000 employees to participate. From these data, we identified sectors with high emotional demands and, based on this, selected and invited workplaces to participate in the project.

Initially, employees and supervisors from 131 workplaces completed a questionnaire one year earlier assessing emotional demands and burnout. A description of the survey administration, timeframe, data collection procedures, questionnaire design, and item selection can be found in a previous report [34]. The survey identified six practical strategies to manage emotional demands, and the results indicated that these strategies were associated with lower burnout levels. Furthermore, higher levels of Psychosocial Safety Climate were statistically significantly associated with the availability of these six practical strategies [35].

For this study, 14 workplaces were selected using a sampling strategy based on baseline data, where the average reported emotional demands were above the median, while the average burnout levels were below the median. This approach was designed to identify workplaces where employees experienced high emotional demands but reported low levels of burnout, indicating the presence of effective preventive strategies.

The participating workplaces included special schools, day centres for individuals with dementia, family and children’s departments in municipalities, job centres, and treatment facilities for substance abusers. We conducted interviews with two groups of employees (with 3–4 individuals each) and one group of supervisors in each workplace. Each group consisted of one to three individuals, except for one group that had six participants. In total, 14 interviews were conducted with supervisors. At three workplaces, there was only one supervisor; at the other locations, there were to 2–3 supervisors who were interviewed as a group. None of the supervisors declined to participate in the study. In total, 31 group interviews were conducted with employees, typically involving to 3–4 employees in each interview. The participants in the employee interviews held various job titles, including teachers, preschool teachers, early childhood educators, social workers, nurses, healthcare assistants, and occupational therapists.

The first and last authors collected data through group interviews, which lasted between 45 and 60 min. Prior to each interview, participants provided informed consent.

The interviews were conducted in the workplace. This choice of location was intended to enhance the ecological validity of the interviews [36]. Conducting interviews in the workplace may enhance recall and accuracy in discussing work-related issues but could also introduce bias or limit openness [37].

We employed an interview guide that included relevant questions aligned with the study’s objectives, ensuring that the scope and direction of the questions were appropriately tailored. Probes were included to elicit additional information, when necessary. The initial question focused on situations where supervisors identified themselves as emotionally demanding for their employees. We examined why these situations are emotionally challenging and what emotions, both positive and negative, are evoked in employees before, during, and after these experiences.

In interviews with supervisors, the theme was introduced through open-ended questions, such as “How do your employees manage high emotional demands? What practice do you take to assist employees in managing these emotional demands?” Each theme included follow-up questions designed to encourage detailed responses and to avoid simple “yes” or “no” answers, such as “What does it mean?” and “Can you provide an example?”

In the interviews with the employees, we asked whether employees perceive that supervisors provide support and assistance that the supervisors themselves assert they offer. The theme was introduced through open-ended questions, such as, “Can you describe a situation involving emotional demands?” and “How does your supervisor help employees manage emotional demands? What specific actions or practice do they take to support employees?”

Questions were developed to allow respondents to convey their own stories in their own terms and to be unobtrusive and nondirective [38]. The interviews were conducted in a semi-structured format centred on a framework of themes rather than a fixed set of questions. The themes included situations with high emotional demands, methods for supporting employees in utilising practical strategies to manage emotional demands, ways to communicate attentiveness and availability to employees, and strategies for establishing a trusting and collaborative organisational climate.

### 2.2. Data Analysis

All interviews were transcribed verbatim, and the resulting transcriptions underwent content analysis following the methodology outlined by Graneheim and Lundman [39]. Familiarity with the data was established through a thorough reading and rereading of the material. Subsequently, NVivo 14 (QSR International Pty Ltd., Version 9.2, 2013) was used to facilitate data coding and categorisation. The analysis process involved coding to identify the variations, similarities, and associations within the material.

The coding and categorisation processes were labelled systematically and abstracted during the analysis. This involved organising data relevant to each code and consolidating these codes into potential themes. The coding approach used was inductive. Through an open-coding process, practices related to supervisory support were identified. The main categories identified included situations characterised by high emotional demand and how the supervisor in practice supported the employees informally and formally within colleague groups and at the organisational level. Preliminary analysis was conducted by an anonymous research assistant. To enhance intercoder reliability, coding was collaboratively discussed and refined by all authors.

## 3. Results

How do supervisors support employees in managing the emotional demands of their work?

Regarding the first research question, “How do supervisors support employees in managing the emotional demands of their work?”, the supervisors mentioned that they supported a range of different practices to support employees in managing emotional demands, such as team meetings and professional sparring, venting, and rotations to exchange cases among team members, being available to provide a quick response, and protecting the employees. In most workplaces, there was also access to formal professional supervision conducted by external psychologists, typically 4–6 times a year.

We illustrate the various practices in the following sections. First, within each type of practice, we present quotes from supervisors describing the practice implemented in the workplace and how they support them. This is followed by quotes from employees at the same workplace, illustrating their perceptions of supervisors’ actions.

### 3.1. Team Meeting and Professional Sparring

At all the workplaces, supervisors organise work in a manner that facilitates regular team meetings and professional sparring sessions. These gatherings allow employees to exchange ideas and receive support from supervisors and colleagues, when needed. Supervisors indicated that through sparring, employees can share knowledge and experiences, which can significantly contribute to their professional development. This is illustrated in the following example, in which the supervisor explains how to manage situations that require further discussion:

We will address this issue in a team meeting. If there is something that is general, or something everyone can learn from, then we will bring it up at the team meeting without exposing anyone. I have employees who say, “I actually want to stand up and share something. I made a huge mistake, but I want to share it with the team”, because it is something we can all learn from, so it is very beneficial in terms of learning; it is something we can develop from and learn from.(Supervisor, job centre)

In this quote, the supervisor expresses the practices of openness, reflectiveness, and collaborative learning within the team. When supervisors encourage employees to voluntarily share their experiences, including mistakes, without fear of judgement, they try to foster a supportive environment that prioritises continuous learning. By discussing challenges in a collective setting, team members can benefit from shared insights, avoid similar errors, and contribute to accountability and personal development.

The supervisors explained that they could see things from the outside. In this way, supervisors can notice whether everyone participates in professional discussions and watch for signs that employees avoid raising issues, which may indicate that something is wrong. Supervisors believe it is important to provide employees with opportunities for reflective practice, which fosters learning and allows them to pause and reflect on how the job affects them.

In the workplace where the previously cited supervisor was in charge, many employees visited citizens individually. It was important for the supervisor at this workplace that employees were not left alone in facing challenges and problems. She explains

[…] When you (the employee) are working alone, there are certain things you do not notice. No matter how skilled you are, you become biased or unable to see yourself from an external perspective. Over time, you also get to know the citizen so well that you sometimes think you know everything: “Well, I know what this is about”, and so you stop asking questions—not out of bad intentions, of course. That’s why I think team meetings are important. It is essential to talk about citizens that you do not normally bring up. It allows you to nudge things a bit and say: “You never talk about this citizen; could you share a few words?” And then sometimes we discover that the process is actually going so well that it could be concluded. Or something comes to light, and as people share, they themselves might realize: “Okay, yes, I might actually be a bit affected, or I am…”(Supervisor, home counsellors)

This quote emphasises that supervisors value consideration, introspection, and reflection in the workplace. The supervisor understands that employees may develop blind spots or assumptions, making it more challenging to notice critical details or recognise personal biases. Team meetings and professional sparring create opportunities to share perspectives, revisit cases, and spark discussions that may not otherwise occur. These meetings can generate new insights, helping employees strengthen their emotional awareness and manage their emotional demands.

Turning to the employees as home counsellors, they largely corroborated the supervisors’ accounts. The employees confirmed that their supervisors provided opportunities for team meetings and professional development. This can be illustrated as follows:

Yes, and then we have our team meetings, where we sit together with the supervisor and the entire team. In my experience, these meetings are not typically for addressing immediate or urgent issues unless it is something that really should have been handled already. They are more suited for reflection, such as recognising patterns after repeated visits to a particular client and acknowledging recurring feelings. It is certainly a setting in which cases can be openly discussed.(Employee, home counsellor)

This quote from the employee highlights the structured support provided through team meetings, demonstrating how employees feel that these meetings serve as a space for thoughtful discussion rather than immediate problem-solving. Employees believe it is valuable to create forums where they can openly share experiences and collaboratively develop strategies to address recurring challenges. Such discussions not only foster collective problem-solving but also provide insight into their own emotions and emotional awareness, helping them better understand and manage emotional demands in the workplace

### 3.2. Venting—Expressing One’s Feelings, Frustrations, or Emotions

At all the workplaces, both formal and informal opportunities are provided to express emotions, often called venting or letting off steam. This could occur during regular team meetings, where time was specifically allocated for this purpose, or through an understanding that employees could approach a “ventilation buddy” or venting could take place in a brief meeting before the end of the workday, allowing employees the chance to express what had been on their minds throughout the day:

And the important thing about overlap, or ventilation at the end of the day. From quarter to three until three o’clock, we withdraw. Then, we sit together and say, ‘How has your day been today?’ What made it good? Or: ‘What made it difficult?’ And it can be something like, ‘I think it has been a good day because I managed to get Paul to do this and that’. Or someone might say something like, ‘Damn, I almost lost my professionalism with John’. So, we talk about what has filled our day. Then, we put our work aside, and we go home.(Supervisor, residence)

This quote highlights the supervisor’s emphasis on a formal setting for emotional expression or to let off steam. By formalising this approach, the supervisor integrates daily overlap sessions, allowing team members to decompress by sharing successes and challenges. This practice not only fosters emotional support, but also enhances emotional awareness, helping employees recognise and process their own emotions.

Employees at the same workplace largely supported the supervisors’ accounts, as further demonstrated in the following:

And that is something we have realised we can talk about. For example: “I can see something is happening for you because I just mentioned that John (the client) needs to have his diaper changed. How does that make you feel?” This way, you can discuss it by making the issue a problem, rather than becoming something personal. This helps to lift it out of the personal realm.(Employee, residence)

This quote highlights that a clear process is in place to make explicit the emotions that have been activated, thus enhancing emotional awareness, which can influence how these emotions are processed. By focusing on discussions of objective situations rather than personal judgments, the team can address challenges in a constructive manner. This approach not only helps normalise difficult emotions but also promotes a professional and supportive atmosphere.

In another workplace, the supervisor described how they had a round in which employees could express what they were preoccupied with. A supervisor explains

We only meet once a week for two hours, which is not a lot, but despite that, we actually spend half an hour each time having a round where we discuss what we are preoccupied with. It can be many things; it can also be something from our personal lives, but it can also be work-related, and it can be something that emotionally affects us… so it’s a round where no comments are allowed.(Supervisor, special unit for heroin treatment)

This quote highlights the importance of supervisors dedicating time for personal and emotional expression in the workplace, even under time constraints. Despite a busy schedule, the supervisor creates a non-judgmental and supportive environment in which employees are encouraged to share their thoughts and emotions openly, thus fostering trust within the team. A key element of this approach is the rule that others should not comment on individual contributions, which may help establish a judgement-free space in the team. This practice may primarily function as training in emotional awareness—encouraging individuals to recognise and articulate their emotions may make it easier to manage them.

Turning to employees from the same workplace, they confirm that employees feel safe to express their thoughts and feelings. This is illustrated as follows:

Employee 1: We have always had fun, enjoyed ourselves, and looked out for each other. We meet privately, go out for beers, and are serious and professional here at work. I think this makes a significant difference. Feeling safe, knowing it’s okay to be upset, taking a day off, or speaking openly about how you are feeling—nobody feels like they must hold back out of embarrassment.Employee 2: I think that it really makes a difference. We genuinely like each other, so we check in: “Is everything okay?” There’s no drama.Employee 1: Being listened to, seen, and heard—regardless of experience or how long you have been here—I think that really creates a solid foundation for mutual trust.(Employee, special unit for heroin treatment)

This quote highlights that a supportive environment at this workplace, where employees feel safe to express their emotions, take time off, and speak openly without judgement, fosters trust. The team’s mutual care and inclusive recognition may strengthen their relationships and promote a sense of belonging, contributing to a positive and supportive work atmosphere.

### 3.3. Discussing Emotionally Demanding Situations

Supervisors were attentive to being available when employees needed to express their feelings or frustrations in high-pressure situations and discuss emotionally demanding situations. However, supervisors have consistently focused on addressing the root causes of these frustrations. By asking reflective questions and encouraging employees to think beyond their immediate concerns, supervisors ensured that the expression of emotions resulted in constructive outcomes rather than mere complaints. They did not simply validate employees’ frustrations; instead, they posed thoughtful questions to prevent frustration from becoming the dominant narrative. This approach is illustrated in the following example:

If an employee comes to me and says, ‘Now you just have to listen, we have a relative who is just really annoying, and when she calls, she’s just annoying’. Then it becomes the only truth about this person, and rarely anything good comes out of it… so we must have the capacity to get out of that narrative again and see why she is annoying?… if not, the employee will be reinforced in the belief that this person is annoying because ‘my manager actually thinks so too, so I must be right’. It then becomes difficult to get out of that mindset.(Supervisor, daycare for citizens with dementia)

This quote emphasises that reinforcing an employee’s one-sided perception of a situation can entrench negative biases, making it challenging for them to adopt a more balanced or empathetic viewpoint. While expressing emotions can yield numerous positive effects, such as fostering a deeper understanding of one’s own needs and emotional awareness, it may also confine employees to negative perceptions of their circumstances. Merely allowing employees to vent without addressing underlying issues can create a cycle of negativity, where problems are discussed but never resolved, potentially resulting in a toxic work environment. The supervisor recognised the importance of helping employees step back from these limited perspectives and explore the broader context, thereby fostering a more open and constructive mindset.

Turning to employees from the same workplace, they largely corroborated the supervisors’ accounts. This is also confirmed by employees as follows:

This was not something that was simply ignored or kept inside. We let off steam, brought in more sparring partners, and vented and looked at it from several different angles. That was really all that it took. Overall, we have a supervisor who is understanding when we come with issues. When we bring up problems, something is done about them, and action is taken to address them. In this particular situation, we brought in help from outside, and in other situations, other options could be aired and tested.(Employee, daycare for citizens with dementia)

From this quote, we can see that employees perceive open communication and a supportive work environment as key elements for managing emotional demands in the workplace. The process of “venting” allowed them to obtain clarity and identify solutions. This underscores the importance of both emotional support and practical action for supporting employees.

### 3.4. Rotation

In many workplaces, there is an opportunity to rotate between the most emotionally demanding tasks to mitigate the impact of such situations. This task rotation strategy not only helps alleviate the psychological strain associated with prolonged engagement in emotionally demanding situations but also promotes a more sustainable work environment. In some cases, the tasks could be ‘diluted’, indicating that employees maintain contact with their clients or duties but at a reduced intensity.

Additionally, there were instances in which employees could be temporarily relieved from their most challenging tasks for shorter durations. In some workplaces, the complete rotation of responsibilities was also an option if the initial approach proved ineffective. This is illustrated in the following example:

You can have contact with citizens where there is simply not the right chemistry, so you should not just keep fighting with each other if it does not yield a result. It is fine to make a change, and it is not because you are failing, no, then we say, ‘Would it be a good idea to make a change here? Are you completely stuck?’ But they also often come in and say, ‘This one, it’s not working’, well then we will figure it out.(Supervisor, job centre)

This quote highlights the importance of recognising when a situation is not working and is open to making changes. The supervisor emphasises that it is not a failure to acknowledge when a particular dynamic, such as the relationship between an employee and a citizen, is not working. Instead of continuing a potentially unproductive or challenging interaction, it is seen as a positive approach to reassess and rotate.

Turning to the employees, they largely corroborated the supervisors’ accounts. This is also confirmed by the employees, as shown in the following:

Employee: When it somehow starts to become too much with that particular client, it is definitely a huge help that we have a supervisor who… there are not many questions asked about it. If we come and say, ‘Now we need to switch’, then a change is made.Interviewer: Is a change made to the client?Employee: Yes, and it’s not because you cannot get help to understand what caused it, with supervision and such, but there is this acknowledgement that yes, working with people does have an impact on us.(Employee, job centre)

This quote emphasises the importance of a supportive and empathetic leadership style in managing challenging situations. The supervisor is portrayed as flexible and responsive, readily accommodating requests for change without questioning or inducing feelings of guilt among the staff. This approach acknowledges that working with clients can be emotionally taxing, and taking proactive measures, such as adjusting the personnel involved, is a constructive way to mitigate emotional strain. Flexibility in managing workloads, such as implementing rotations when necessary, can foster supportive work environments.

### 3.5. Being Available to Provide Prompt Feedback

The supervisors typically emphasised their availability to address issues promptly, ensuring that employees did not have to navigate challenges independently and without support. They demonstrated a willingness to prioritise employee support by temporarily setting aside their own tasks to provide assistance. They made a point of being available to provide quick responses to issues, so that employees would not have to deal with problems on their own.

This can be illustrated as follows:

Yes, but if you just need to say something to me, if you just need to take a quick breath and say, ‘Oh no!’ and then continue, so they feel free to do so… and sometimes people can be under pressure, and you can see the tears in their eyes, then I put down everything I have in my hands, and save my work after 4 p.m. That’s what I do.(Supervisor, home counsellors)

I also think that openness—the fact that we are open about how tough it is to work here —gives counsellors a sense of safety to dare to come in and ‘throw up’, so to speak; that’s what we call here. Meaning, to come to their supervisor and say, ‘I can’t handle this’, or ‘I’m struggling’, or ‘I simply can’t do it—you have to help me.’ They also feel safe in doing so. It is okay for them to come in and say this. They feel comfortable coming to me and saying it.(Supervisor, family department)

The first quote highlights that supervisors highly value being present and emotionally available to their employees, thereby fostering a supportive environment for processing challenging emotions. The second quote highlights the importance of psychological safety in the workplace. By being transparent about the challenges of the job, the supervisors create an environment where employees feel comfortable expressing their struggles without fear of judgement. This openness makes counsellors feel safe seeking help when they are overwhelmed.

Turning to the employees, they largely corroborated the supervisors’ accounts. They indicated that the supervisors prioritise employee support by temporarily setting aside their own tasks to provide assistance. Employees at the same workplace affirmed this:

Interviewer: What does the supervisor do to support you in managing these high emotional demands?Employee: I think the frameworks that have been established, such as supervision, team meetings, and general openness, are really positive. If there is something serious that needs to be discussed and resolved, I always feel that there is time for it.Interviewer: So, you can… the door is open. You can just come in?Employee: Yes.Interviewer: And what does the supervisor do then?Employee: Well, she takes the time to listen, makes time available if you ask for it, and also provides feedback.(Employee, family department)

The quote illustrates that the employee values the supervisor’s commitment to openness and the “open door” policy. The supervisor’s readiness to “take the time to listen” and “make time available” underscores a vital aspect of emotional support—being present and accessible when needed. Additionally, the supervisor’s provision of feedback is another essential element. This demonstrates that, beyond merely listening, the supervisor actively engages with employees and offers feedback, thereby reinforcing a sense of support and resolution.

### 3.6. Safeguarding Employees’ Well-Being in Emotionally Demanding Situations

Several supervisors shared their efforts to safeguard employees’ well-being by managing their exposure to emotionally demanding situations. This support could involve shielding employees from high-intensity interactions, such as those involving collaborative challenges with demanding relatives or clients, or monitoring situations where an employee’s strong engagement in their work leads to excessive dedication, potentially at the expense of personal boundaries:

When there are issues such as with a relative, there are boundaries regarding the extent of involvement that employees should have in certain situations. In such cases, we intervene by advising employees to refer the relative to us if the situation escalates. We then handle the conversation with a relative. This approach allows employees to maintain positive relationships with their relatives for as long as possible while providing support and assurance that they have our support.(Supervisor, daycare for citizens with dementia)

This quote emphasises that the supervisor recognises the importance of establishing clear boundaries and providing support to employees in sensitive situations involving family members. By intervening to manage difficult conversations when necessary, the supervisor enables employees to maintain positive relationships while feeling supported.

Additionally, employees may experience fewer interactions, which can increase the risk of ambiguous emotions. When shielded from interpersonal conflicts, such as those involving relatives, they may encounter fewer frustrating emotions. In turn, this allows them to focus more fully on the emotions that arise in their work with citizens or clients.

By reducing the exposure to emotionally demanding situations related to conflicts with relatives, employees may be better equipped to engage empathetically and professionally, ultimately enhancing the quality of their interactions and the support they provide. 

Turning to the employees, they largely corroborated the supervisors’ accounts. This is expressed as follows:

[…] Sometimes, there is a relative on the other end of the phone who is frustrated, and if things get stuck in a deadlock—well, it can happen that for one reason or another, a relative can no longer communicate with us. In those cases, our supervisor is really good at stepping in and saying, “Alright, we’ll take it from here”.(Employee, daycare for citizens with dementia)

This underscores the importance of supervisory support in managing challenging interactions. When communication with relatives becomes difficult or reaches an impasse, the supervisor intervenes to take control, thereby ensuring that the situation is handled professionally. The supervisor’s readiness to step in exemplifies proactive leadership and guarantees that employees are not left to navigate complex or frustrating interactions on their own. This support is greatly appreciated by employees.

It is not always that straightforward. 

So far, we have observed examples of practices in which supervisors are proactive and take initiative. However, the situation is not as straightforward as it seems initially. Supervisors may be uncertain whether to adopt a proactive stance or take a more wait-and-see approach. Several supervisors shared their uncertainty, which is evident in the following example:

The youth team encountered a situation in which one member expressed frustration over the lack of support from management. She stated, “I want you to come and ask me after I’ve been in situations like that, and then I want you to follow up the next day, something like that”. I responded, “I didn’t know that because I don’t know you very well”. This highlights the dynamics we face as substitutes; does it not? This is a clear example of this. She clearly had an expectation that I should be proactive, but not everyone shares that way. For some, if you are overly proactive with others, they might find it somewhat bothersome, asking’, Why are you constantly checking in? that’s true. It’s a balance. It’s a very fine balance to maintain.(Supervisor, job centre)

This quote underscores the supervisor’s experience that employees have varying expectations regarding the support they desire. Some may appreciate proactive engagement, whereas others may perceive it as intrusive. This supervisor finds it challenging to recognise varying expectations and tailor interactions accordingly. Furthermore, for temporary supervisors, navigating interpersonal dynamics can be difficult because of their lack of familiarity with team members. This situation underscores the complexity of interpersonal relationships in professional settings and the necessity for thoughtful and individualised approaches to support and communication. Each employee brings unique experiences, emotions, and needs, which means that a one-size-fits-all approach may be ineffective. This requires not only an awareness of the emotional dynamics at play but also the flexibility to adapt their approach based on the specific circumstances and individuals involved.

At the same workplace, employees expressed high levels of satisfaction with their supervisor’s support, but there was also a group of dissatisfied employees. This dissatisfaction is evident in the following example:

Employee 1: What we may lack in terms of leadership is a greater degree of proactivity. Our supervisors are aware when you are involved in a challenging case, facing a difficult situation, or dealing with personal struggles. Sometimes you may wish for more support in these instances. We have come to recognise that this level of engagement is absent from our current supervisor. However, we acknowledge and accept this reality, at least in my opinion.Employee 2: At least we have acknowledged it.Employee 2: Yes, we acknowledge this, and we support each other. I wish that our supervisor—having experienced various supervisors during my time here—was proactive in checking in. For instance, it would be great if the current leadership would pop in and say, “Hey, that case you mentioned last week was pretty intense. How have you been doing since then?”(Employees, job centre)

From this quote, it can be seen that employees value supervisors who take the initiative to check in, especially in challenging situations. They appreciated when their supervisors demonstrated genuine concern for their well-being by following up on specific issues or events, indicating a more personalised approach from their supervisors. However, both employees appeared to have accepted the current limitations in leadership while expressing a desire for change. In the absence of strong supervisory support, the employees often rely on each other for emotional and professional support.

### 3.7. Constraints on Supervisor Support

Although supervisors make significant efforts to help employees manage the high emotional demands of their work, there are structural factors beyond their control. One supervisor described a large caseload, many clients, extensive documentation requirements, and time pressures. She explained

This year, we went through two rounds of budget cuts, and it has left its mark. Not only because we are fewer people handling essentially the same tasks, but also because it takes a toll on everyone involved…. The money has gone, and savings must be made. So, one thing is the increased workload, but another is the growing bureaucracy. There is more documentation to complete, more requirements are added to our tasks, and on top of that, there is a fear of losing one’s job. We try to mitigate the effects as much as possible—we make arrangements such as temporarily reducing caseloads or exempting employees from certain tasks. We do everything that we can support our staff and provide a safety net. However, at the end of the day, we are working with people, and they also have personal struggles—whether financial difficulties, trouble paying electricity bills, or other personal challenges—adding to the burden beyond just the job itself.(Supervisor, job centre)

This quote illustrates that the supervisor acknowledges that she does not have full control over all the factors influencing employees’ emotional demands, such as budget cuts, bureaucratic requirements, and burdens beyond the job itself.

The supervisor describes a complex situation in which workload pressures, documentation demands, and a fear of job loss all contribute to employee strain. While efforts are made to support employees, external factors further compound pressure, highlighting the limits of organisational support in mitigating these challenges. The supervisor implicitly recognises that there are constraints on what they can do to provide support.

How do supervisors build and maintain trust to enhance the effectiveness of their support in helping employees manage the emotional demands?

Regarding the second research question, ‘How do supervisors build and maintain trust to enhance the effectiveness of their support in helping employees manage the emotional demands?’, the supervisors stated that they build and maintain trust by being accessible, perceiving mistakes as learning opportunities, and trusting employees.

First, for each topic, we present quotes in which supervisors describe their actions. This will be followed by quotes from employees at the same workplaces, illustrating their perceptions of the supervisors’ actions.

It was characteristic that the supervisors highlighted the importance of fostering an environment where employees could openly share their thoughts, feelings, concerns, and ideas without fear of judgement or ridicule. Many supervisors have underscored the importance of developing a climate of trust that encourages mutual support among team members. They aimed to create a workspace in which employees felt secure in taking interpersonal risks and free from the fear of negative repercussions. Supervisors were aware of their critical role in shaping a trusting relationship with employees and actively sought to enhance such conditions to support employee well-being and engagement.

The points will be elaborated on in the following section.

### 3.8. Being Accessible

Many supervisors were highly attentive to being accessible. This was often demonstrated by keeping their doors open and maintaining a physical presence near their employees, which signalled a culture of openness. The supervisors recognised the importance of allowing employees to easily share their ideas, feelings, and concerns, thereby facilitating prompt guidance. One supervisor mentioned that she intentionally leaves her door open; when employees notice it “…they come in and actually just talk about this and that….” (Employee, family department).

One supervisor explained the following:

I think what we do most of all is to be available. It is like having a bit of a container function, where they can come to us and unload when everything becomes too much. They use us for that. We always have time available, so they can just have a moment to talk about their day, say what has been really good, and what has not been so good today.(Supervisor, daycare for citizens with dementia)

This quote shows that the supervisor recognises the importance of close, accessible, and emotionally engaged leadership in supporting employees. The quote illustrates the supervisor’s belief in their ability to provide a “container function” by creating a safe space for employees to process their emotions, reflect on their day, and discuss both positive and challenging experiences. These quotes highlight the significant value of supervisors being available, attentive, and emotionally supportive.

Turning to the employees, they largely corroborated the supervisors’ accounts. This is shown in the following:

Employee 1: I also think it’s about having open leadership. Leadership plays a very important role; it has been here for many years and has just recently changed, but it has always been a significant part of things. You can always go in, be heard, and receive good advice.Employee 2: There’s also recognition.Employee 1: Yes, a lot of recognition and humour No matter how busy it gets, there’s always been time for the little things.(Employees, daycare for citizens with dementia)

These quotes illustrate that employees appreciate having a supervisor who is accessible, supportive, and engaging. They highlight the importance of an accessible supervisor who listens to, offers guidance, and acknowledges employees’ contributions. The mention of humour indicates that positive and relaxed relationships between employees and supervisors are also highly valued, even during busy periods.

### 3.9. Mistakes as Learning Opportunities

The supervisors acknowledged that mistakes are a natural part of the learning process. They emphasised that employees should not fear making mistakes as they will not face punishment. Instead, mistakes can serve as valuable opportunities for organisational learning:

We may have complaints about the employees, and I always take them and say, ‘Well, listen, I have got your back. So, no matter what happens, I am not going to throw you under the bus or anything. And if someone has to be thrown under the bus, I’ll jump with you.’ If a mistake has been made, a mistake has been made. Hopefully, we have become wiser and will do things differently next time. And of course, there are times when we can’t, when we have to say that mistakes have been made, yes. But then we talk about it, and there is no blame or shame.(Supervisor, family centre)

In the quote, the supervisor expresses a supportive and protective leadership style, through which she assures her team that they will not be abandoned or blamed for mistakes. Instead, the emphasis is on accountability, reflection, and the identification of ways to improve in the future. The supervisor emphasises that there should be no shame in making mistakes and that open discussions will help to foster growth and enhance decision-making.

Taken together, the supervisor’s statements reflect a leadership philosophy that prioritises trust, emotional safety, and continuous learning in the workplace.

Turning to the employees, they largely corroborated the supervisors’ accounts. This is shown in the following:

The supervisors have a deep understanding of what we’re dealing with. They are informed when people file complaints, and they also have to meet with those individuals. So, they are fully aware of the situation we face and understand that there are parents who are constantly making noise…. And it’s nice to have support from the supervisor knowing that what I say is right.(Employee, family centre)

This quote suggests that employees believe that their supervisors possess a deep understanding of the challenges they encounter. Employees value supervisors who not only acknowledge these difficulties but also validate their experiences and perspectives. The assurance of support from their supervisors instils confidence in employees, affirming that their concerns and actions are justified, even when faced with complaints from parents. This dynamic fosters a sense of trust and support in the workplace.

### 3.10. Trust in Employees

Some supervisors encourage their employees to express themselves freely and utilise their abilities without fear of facing detrimental repercussions on their self-esteem, reputation, or professional advancement. A supervisor explains:

I think it is incredibly important to have trust in each other, so much trust that you can freely say anything—in relation to how it affects you and how vulnerable you also become yourself. Even though I have been in the field for many years, sometimes I can think, ‘I thought I had heard or seen the worst, and now this came’. We constantly try to have an open space where we can talk and where we trust each other, and where one does not look critically at someone because they are emotionally affected. Some are affected by one thing, whereas others are affected by something completely different. And we must have respect for and an understanding that we are different as individuals.(Supervisor, crisis centre for women affected by violence)

In this statement, the supervisor underscores the importance of building a workplace culture based on trust, openness, and emotional support. This shows that the supervisor recognises the need for employees to feel safe sharing their feelings and vulnerabilities without fear of criticism. She also stresses the importance of recognising and respecting individual differences, as people respond to emotional situations in unique ways. By fostering open communication and a judgement-free environment, the supervisor tries to create a supportive atmosphere where employees feel valued and better equipped to manage the emotional complexities of their roles.

Turning to the employees, they largely corroborated the supervisors’ accounts. This is shown in the following:

Employee 1: So, there’s always backup, and we’re allowed to, we can handle it. There is respect for the fact that we are dealing with it.Employee 2: Yes, leadership that is built on trust in the employees—it just works well. Trust and a sense of safety. These are the keywords, right? Safety and room for vulnerability. Yes, this is important.(Employee, crisis centre for women affected by violence)

The quote demonstrates the employees’ experience of a workplace culture built on trust, respect, and support. Employees feel empowered to address challenges independently and are confident that they have reliable backup when needed. Trust and psychological safety appear to be essential components for maintaining a supportive environment.

## 4. Discussion

This study aimed to gain a deeper understanding of how supervisors support employees in managing emotional demands at work and how they build and maintain trust within the workplace. Despite the structural differences, the participating workplaces were highly comparable, likely due to their selection based on the shared characteristics of high emotional demands and low burnout rates.

Our findings indicate that supervisors use a variety of informal and formal practices designed to assist employees in managing emotional demands in the workplace. These practices include the opportunity for supervision; the opportunity to discuss emotionally demanding patients, citizens, or students; the opportunity to provide prompt feedback; the opportunity to “vent” with colleagues or management; the opportunity to rotate tasks; and the opportunity to discuss strategies for managing high emotional demands.

Most workplaces offer access to professional supervision, although this typically occurs only every four to six weeks. Supporting a variety of informal and formal practices to manage emotional demands was just one aspect of the supervisors’ approach to supporting their employees. Another aspect involved emphasising the importance of cultivating a climate of trust, where supervisors prioritised creating a work environment in which employees felt secure enough to take interpersonal risks without fear of negative consequences. They achieved this by being accessible, viewing mistakes as learning opportunities, and placing trust in their employees.

These two approaches were intertwined and complementary. It is impossible to assert that one was a prerequisite for the other; rather, both approaches should be regarded as mutually supportive.

There was a consensus among the workplaces regarding the usefulness of applying both formal and informal practices. However, variations emerged in the extent to which these practices were formalised. Additionally, some supervisors struggled to provide the expected support, leading to employee dissatisfaction. When such issues occur in best-practice workplaces, this may indicate that there may be even more significant support challenges in workplaces that are not considered best-practice environments.

### 4.1. Interpretation

Several studies have found positive associations between supervisor support and the better management of emotional demands [16,17,18]. However, the effects of supervisor support remain unclear. However, the key lies in effectively translating the findings into practical guidance [40]. This study emphasises a particular practice that supervisors can adopt to support employees in managing their emotional demands. Supervisors in this study describe the particular practices they employ to support employees in this context, including team meetings, professional feedback, task rotation, prompt responsiveness, and opportunities for emotional venting.

By providing feedback and opportunities for sparring, supervisors help employees improve their skills, which can boost their confidence and sense of mastery, and by opening dialogue about challenging situations, supervisors create a culture where it is acceptable to talk about challenges and emotions. Previous studies found that it can be beneficial for employees to receive feedback and support from their supervisors [41,42]. Research supports this approach, with meta-analytic findings indicating that feedback behaviours are positively correlated with job satisfaction and proactive behaviours [41]. Rotating tasks can provide employees with a respite from demanding or emotionally taxing responsibilities, thereby helping to prevent burnout. A meta-analysis found that task rotation is linked to enhanced learning and development and improved psychological health [43]. A commonly used initiative was venting. Although venting can offer temporary emotional relief, it is generally considered a maladaptive coping mechanism [44]. It is important for employees to feel heard and understood [45]; however, it is equally crucial for supervisors to guide conversations towards constructive solutions rather than allowing venting to become habitual complaining [46]. The supervisors seem to be aware of this and highlighted the importance of using reflective questioning when employees engaged in venting as a strategy to manage emotional demands. They noted that simply allowing venting without addressing root issues could perpetuate a cycle of negativity, where problems are aired but remain unresolved, potentially fostering a toxic work environment and intensifying the adverse effects of negative emotions [47,48]. By encouraging employees to reflect on their frustrations and consider constructive perspectives, supervisors sought to ensure that emotional expression led to productive outcomes, rather than contributing to complaints or grievances.

All the practices of the supervisors may help employees maintain a healthy balance between empathy and action, fostering a more supportive work environment. These efforts enable employees to navigate through the complexities of emotionally demanding roles without becoming overwhelmed or distanced. Effectively managing one’s emotions is a critical factor in addressing the emotional demands of their work [14]. Team meetings, rotations, prompt responsiveness, and opportunities for emotional venting implemented by supervisors may assist employees in maintaining their professionalism and empathy by managing their emotions to meet their job requirements. The existing research highlights discrepancies between supervisors’ and employees’ perceptions of support, with supervisors often overestimating their roles [49]. Prior studies have primarily focused on employees’ perspectives [16,17,18]. By contrast, this study includes both viewpoints and, unlike previous findings, reveals a high degree of coherence between them. Our findings suggest that supervisors and employees largely align in their descriptions of the practices and strategies used to manage emotional demands, indicating a shared understanding of supportive practices in the workplace.

Such perceptual congruence between leaders and employees may be associated with a greater leadership effectiveness, as aligned perceptions foster trust, enhance communication, and contribute to improved organisational outcomes [50]. The alignment between supervisors’ and employees’ perceptions of support can mitigate misunderstandings and facilitate effective communication. One explanation for the alignment between supervisors’ descriptions of the initiative and employees’ perceptions may be that the participating workplaces in this study were predominantly small organisations. In smaller workplaces, characterised by closely knit teams and direct communication, there is a greater likelihood of aligned perceptions among employees and supervisors. Fewer hierarchical layers facilitate a shared understanding of organisational goals, thereby enhancing collaboration and cohesion.

The study conducted by Lamothe et al., 2021, highlight that the effectiveness of supervisor support is contingent upon the quality of the relationship between supervisors and employees [19]. In the present study, it was evident that supervisors prioritised the quality of the relationship between supervisors and employees and the creation of an environment in which employees felt trusted. This is in line with previous studies, and a meta-analysis found support for the relationship between psychological safety and positive leader relations as a general category [51]. This finding highlights the importance of direct supervisors in shaping a safe and trustworthy work environment. Establishing a trustful relationship appears to complement the benefits of the practice of supervisory support.

Some supervisors were uncertain whether to take a proactive or reactive approach to supporting their employees. The complexity of interpersonal relationships in professional settings and the necessity for thoughtful, individualised approaches for support and communication place huge demands on supervisors’ empathy skills [52]. Each employee brings unique experiences, emotions, and needs, which means that a one-size-fits-all approach may not be effective. Supervisors must be sensitive and navigate a delicate balance between being proactive and allowing space for employees to voice their concerns when ready [53]. Providing appropriate feedback to employees will give them greater confidence, increase their sense of safety, and make them feel more comfortable putting in extra effort and exploring creative ways to perform their tasks [54]. This requires not only an awareness of the emotional dynamics at play but also the flexibility to adapt their approach based on the specific circumstances and employees involved.

In Denmark, many social and healthcare workplaces have been subjected to efficiency demands, various requirements, cost savings, and budget cuts. Several supervisors spoke about a heavy workload and extensive documentation requirements and budget cuts that may impose significant structural constraints on supervisors’ ability to provide adequate support for their employees. With reduced resources and staffing and increased demands, supervisors may face increased workloads and limited flexibility in supporting employees’ emotional and professional needs [55]. These constraints affect supervisors’ capacity to implement practices to manage emotional demands.

The practical implications of this study establish formal practices, such as supervision; discussions of emotionally demanding patients, citizens, or students; prompt feedback; “venting” with colleagues or management; task rotation; the opportunity to discuss strategies for managing high emotional demands; and protecting employees, which are essential for effectively supporting employees in managing emotional demands. Organisations should prioritise the implementation and integration of these practical strategies to address the varying needs of employees in managing emotional demands. While these practices provide the necessary structure and consistency for support mechanisms, trust in employees plays a critical role in fostering a supportive environment that facilitates their implementation [34]. By employing both formal and informal practices to support employees in managing emotional demands, supervisors must be mindful of cultivating trusting relationships with their employees. This dual approach not only addresses immediate emotional demands but also enhances supportive and collaborative interactions.

### 4.2. Strengths and Limitations

This study had several strengths. By interviewing supervisors and employees from 13 different workplaces across various sectors, this selection offers a diverse and comprehensive sample. This diversity enhances the applicability of the findings as it captures a wide range of experiences and perspectives regarding how supervisors support employees in managing emotional challenges. Furthermore, the unanimous agreement of all supervisors and employees to participate reflects a strong commitment to and interest in the research topic, which may yield detailed and insightful responses. Additionally, interviews were conducted with both supervisors and employees from the same workplace, but separately, enabling the validation of supervisors’ accounts of their own efforts. The voluntary nature of the participation can improve the reliability of the data as participants are more likely to share their genuine feelings and experiences when they are not coerced into participating.

This study mainly utilised group interviews with supervisors and employees because of the numerous benefits of this method. By gathering input from multiple participants, group interviews captured a range of perspectives and insights, resulting in detailed and nuanced explanations. Additionally, the group setting encouraged the sharing of ideas and thoughts that might remain unspoken in one-on-one interviews. Despite these advantages, group interviews have limitations. Social norms and group dynamics can potentially influence participants to conform to the majority opinion or withhold information [56], making it difficult for researchers to address certain issues during interviews.

Despite these strengths, this study had some limitations. One limitation of this study was the selection of the participating workplaces. The workplaces, and consequently the supervisors, were chosen based on a survey indicating high emotional demands in the workplace coupled with a relatively low level of burnout. As a result, the selected workplaces represented the best practices. While this study focuses on managing emotional demands in daily practice, we do not consider this selection an obstacle to the validity of the findings. This research is a best-practice study, and it is crucial to acknowledge this condition when drawing conclusions. There is a risk that supervisors may exaggerate the beneficial practice they claim to be implementing for their staff to please researchers during interviews [57]. However, the employee feedback largely corroborates supervisors’ descriptions, suggesting that supervisors do not exaggerate the beneficial practice they claim to be implementing for their staff. Another limitation of this study is that the supervisors selected the participating employees. Although we aimed for representation from various professional groups and seniority levels, the selection process may have favoured employees with positive perspectives.

A limitation of this study is that only a few critical employees were interviewed. The absence of critical voices in the interviews could be explained by several factors. First, selection bias might have led to the inclusion of best-case workplaces, where leadership and employee relations are particularly strong. Second, social desirability bias could have influenced participants to avoid criticism, even if anonymity was guaranteed. Third, the study’s focus on supportive and best practices may have steered participants toward emphasising the positive aspects. Additionally, it is possible that participants genuinely felt satisfied with their management because of the effective practices observed in these workplaces. Expanding the sample to include more settings could provide a more comprehensive understanding.

Finally, it may be a limitation that our sampling strategy was limited to a demarcation based on the core factors of emotional demands and burnout. It is possible that the inclusion of additional factors relevant to the complex process of burnout—such as the personal meaning employees attach to emotional demands, individual emotional regulation skills, and external positive emotional connections with clients—could have improved the sample. However, the purpose of this study was not to provide a comprehensive understanding of the multifaceted relationship between emotional demands and burnout. Rather, this study aimed to examine how supervisors support employees in managing the emotional demands of their work. Therefore, sampling based on these two central phenomena was considered key to including a relevant yet varied sample of respondents, reflecting the diversity in the workforce while still ensuring the inclusion of meaningful experiences related to the process we aimed to investigate.

## 5. Conclusions

This study provides valuable insights into how supervisors support employees in managing emotional demands at work and fostering trust within the workplace. The findings indicate that supervisors use a combination of formal and informal strategies, including team meetings, professional sparring, emotional venting, task rotation, prompt responses to challenges, and protective measures, to address emotional demands. Equally significant is the role of supervisors in cultivating a climate of trust where employees feel secure in taking interpersonal risks. This can be achieved through accessibility, framing mistakes as opportunities for learning, and demonstrating trust in the employees.

These two strategies, managing emotional demands and fostering trust, are deeply interconnected and complementary. Rather than viewing one as dependent on the other, the results indicate that they should be understood as mutually reinforcing elements of effective supervisory support. Together, these factors contribute to a supportive, resilient, and trustworthy workplace environment.

Based on these findings, it is recommended that supervisors prioritise managing emotional demands and fostering trust in the workplace. Balancing these interconnected strategies will assist employees in effectively managing their emotional demands.

## Data Availability

The data presented in this study are available on request from the corresponding author.

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
