# Peer review of "How Superiors Support Employees to Manage Emotional Demands: A Qualitative Study"

_ijerph, 2025, doi:10.3390/ijerph22050670_

Round 1
Reviewer 1 Report
Comments and Suggestions for Authors
The study is comprehensive, and the methodological approach is appropriate. The use of NVivo 14 for qualitative data analysis is suitable and aligns with standard qualitative research practice.
This study examines the role of supervisory support in managing employees’ emotional demands and fostering workplace trust. As stated in the article’s abstract, emotional demands pose significant health risks, including burnout, depression, and prolonged sickness absence. The article provides multiple quotes addressing key themes such as supervisory support, emotional demands, trust building, and burnout. However, the challenges of “prolonged sickness absence”, which is explicitly mentioned in the abstract, is not adequately addressed in the main body of the article.
Minor Revision:
It is unclear whether the authors did not find relevant data during their survey or deemed it insignificant. If this issue was not identified, the authors should explicitly acknowledge this in their findings for clarity. Addressing this gap would enhance the coherence between the abstract and main discussion, ensuring alignment between the study’s and its findings.
The topic is interesting, some suggestions related to the article are given below.
- It is suggested to clarify how trust-building affects “prolonged sickness absence”, as no quote or detail is provided in the study, although mentioned in the abstract line 11.
- The statement “14 workplaces were sampled” in already mentioned in line 185, making its repetition in line 194 It would be more effective to integrate it with the following sentence, specifying that these 14 workplaces included….
Author Response
Open Review
Dear reviewer 1
We appreciate the time and effort you have dedicated to reviewing our manuscript. Your constructive feedback has helped refine and strengthen our work. We are grateful for your comments and suggestions, which have helped us improve the quality of the manuscript.
Below, we provide a detailed account of the changes made, as well as our responses to specific suggestions.
Thank you once again for your constructive input. We look forward to your response.
( ) I would not like to sign my review report
(x) I would like to sign my review report
Quality of English Language
( ) The English could be improved to more clearly express the research.
(x) The English is fine and does not require any improvement.
|
Yes |
Can be improved |
Must be improved |
Not applicable |
|
|
Does the introduction provide sufficient background and include all relevant references? |
(x) |
( ) |
( ) |
( ) |
|
Is the research design appropriate? |
(x) |
( ) |
( ) |
( ) |
|
Are the methods adequately described? |
( ) |
(x) |
( ) |
( ) |
|
Are the results clearly presented? |
(x) |
( ) |
( ) |
( ) |
|
Are the conclusions supported by the results? |
(x) |
( ) |
( ) |
( ) |
Comments and Suggestions for Authors
The study is comprehensive, and the methodological approach is appropriate. The use of NVivo 14 for qualitative data analysis is suitable and aligns with standard qualitative research practice.
This study examines the role of supervisory support in managing employees’ emotional demands and fostering workplace trust. As stated in the article’s abstract, emotional demands pose significant health risks, including burnout, depression, and prolonged sickness absence. The article provides multiple quotes addressing key themes such as supervisory support, emotional demands, trust building, and burnout. However, the challenges of “prolonged sickness absence”, which is explicitly mentioned in the abstract, is not adequately addressed in the main body of the article.
Our reply: We have changed the first sentence in the abstract to this: Previous research has found that emotional demands in the workplace can be taxing and contribute to an increased risk of mental health challenges, including burnout and, depression.
Minor Revision:
It is unclear whether the authors did not find relevant data during their survey or deemed it insignificant. If this issue was not identified, the authors should explicitly acknowledge this in their findings for clarity. Addressing this gap would enhance the coherence between the abstract and main discussion, ensuring alignment between the study’s and its findings.
Our reply: The quantitative part of the study highlights the relationship between specific strategies and burnout, and these results are reported in another article. Furthermore, the significance of the psychosocial safety climate has also been examined and discussed in another article.
We have added this in the method section on page 8: The survey identified six practical strategies to manage emotional demands, and the results indicated that these strategies were associated with lower burnout levels. Furthermore, higher levels of Psychosocial Safety Climate were statistically significantly associated with the availability of these six practical strategies [35].
The topic is interesting, some suggestions related to the article are given below.
- It is suggested to clarify how trust-building affects “prolonged sickness absence”, as no quote or detail is provided in the study, although mentioned in the abstract line 11.
Our reply: As mentioned above, we have changed the first sentence
- The statement “14 workplaces were sampled” in already mentioned in line 185, making its repetition in line 194 It would be more effective to integrate it with the following sentence, specifying that these 14 workplaces included….
Our reply: We have changed the sentence for clarity to this on page 8: Initially, employees and supervisors from 131 workplaces completed a questionnaire one year earlier assessing emotional demands and burnout. A description of the survey administration, timeframe, data collection procedures, questionnaire design, and item selection can be found in a previous report [34]. The survey identified six practical strategies to manage emotional demands, and the results indicated that these strategies were associated with lower burnout levels. Furthermore, higher levels of Psychosocial Safety Climate were statistically significantly associated with the availability of these six practical strategies [35].
Reviewer 2 Report
Comments and Suggestions for Authors
Thank you for the opportunity to review this manuscript.
In sections 1.1 and 1.3 (Introduction), the concept of emotional demand requires further elaboration. Prior to discussing the research purpose (p. 2, lines 50-53) and research gap (p. 3, lines 101-132), this study only reviews a single previous investigation on the relationship between superiors' support and emotional demands using a qualitative approach. However, numerous other studies have explored similar topics, which should be incorporated into the literature review. Furthermore, this study should explicitly identify the specific workplace contexts under examination and provide a clear rationale for why these particular workplace environments merit investigation as research objects.
In Section 2 (Methods), when describing the collection of non-numerical data, the survey methodology should be explicitly detailed, including the specific survey administration timeframe, data collection procedures, and a thorough justification of the questionnaire design and item selection.
In the section 3 (Data collection), I cannot find the baseline data of the survey depending on the research of “Andersen, L.P., Andersen, D. R. Pihl-Thingvad, Jesper, Reducing the Risk of Burnout: Identifying Practical Strategies to Manage Emotional Demands - A Mixed Methods Study. 2024”. Additionally, while the pilot survey aims to 'identify workplaces where employees experienced high emotional demands but reported low levels of burnout,' this approach overlooks the complex, counterintuitive, and nonlinear relationship between emotional demands and burnout. The analysis should account for factors beyond internal workplace resources, including the personal meaning employees attach to emotional demands, individual emotional regulation skills, and external positive emotional connections with clients. These additional variables should be either considered or controlled for in the analytical process to provide a more comprehensive understanding of this multifaceted relationship.
Comments on the Quality of English LanguageThere are grammatical problems in this study. For example, the sentence 'Therefore, this study aims to explore the practice though which supervisory support can help employee managing emotional demands' (p. 2, line 52) should be corrected to: 'Therefore, this study aims to explore the practices through which supervisory support can help employees manage emotional demands
Author Response
Open Review
Dear reviewer 2
We appreciate the time and effort you have dedicated to reviewing our manuscript. Your constructive feedback has helped refine and strengthen our work. We are grateful for your comments and suggestions, which have helped us improve the quality of the manuscript.
Below, we provide a detailed account of the changes made, as well as our responses to specific suggestions.
Thank you once again for your constructive input. We look forward to your response.
(x) I would not like to sign my review report
( ) I would like to sign my review report
Quality of English Language
(x) The English could be improved to more clearly express the research.
( ) The English is fine and does not require any improvement.
|
Yes |
Can be improved |
Must be improved |
Not applicable |
|
|
Does the introduction provide sufficient background and include all relevant references? |
( ) |
( ) |
(x) |
( ) |
|
Is the research design appropriate? |
( ) |
( ) |
(x) |
( ) |
|
Are the methods adequately described? |
( ) |
( ) |
(x) |
( ) |
|
Are the results clearly presented? |
( ) |
(x) |
( ) |
( ) |
|
Are the conclusions supported by the results? |
( ) |
(x) |
( ) |
( ) |
Comments and Suggestions for Authors
Thank you for the opportunity to review this manuscript.
In sections 1.1 and 1.3 (Introduction), the concept of emotional demand requires further elaboration.
Our reply: We have expanded the first paragraph as follows: Professions such as nursing, medicine, social work, teaching, and various service-oriented sectors carry significant responsibilities concerning health, well-being, and overall support of patients, clients, students, and consumers. Employees in these fields are often tasked with engaging individuals who experience traumatic or stressful life events, which exposes them to significant emotional demands [1]. A defining characteristic of these professions is the necessity for sustained social interaction, which requires employees not only to engage with those they serve in a professional capacity, but also to manage the emotions expressed by individuals in distress [2]. This interaction creates emotional demands, especially when engaging with individuals facing severe illness, accidents, grief, crises, or other social adversities, which collectively contribute to the emotional burden experienced by professionals in these roles [3, 4].
Prior to discussing the research purpose (p. 2, lines 50-53) and research gap (p. 3, lines 101-132), this study only reviews a single previous investigation on the relationship between superiors' support and emotional demands using a qualitative approach. However, numerous other studies have explored similar topics, which should be incorporated into the literature review.
Our reply: We agree that a broader inclusion of qualitative studies exploring the relationship between supervisor support and emotional demands is important. Our aim in this section has been to identify studies that specifically examine, from the supervisor’s perspective, how they perceive their role in supporting employees in relation to, for example, stress and burnout. However, it has proven challenging to locate studies that address this issue from the supervisor's point of view. We have conducted several additional searches in collaboration with our research librarian and have now identified and included several relevant studies in the revised introduction.
First we have added this on page 5:
This gap can make it challenging to establish consistent guidelines or actionable steps, ultimately complicating efforts to translate interventions aimed at improving supervisor support within organizational settings.
Despite these challenges, qualitative studies have highlighted the importance of such efforts. For instance, a comprehensive review of 13 qualitative studies underscores the necessity and significance of implementing support and supervision mechanisms to assist staff in managing diverse emotional challenges associated with their professional responsibilities [1]. Other qualitative studies have investigated which factors employees perceive as being associated with health-promoting leadership and supervisory strategies for managing occupational stress and returning to work. For instance, interviews were conducted with 39 employees who had been absent from work for at least three weeks. The findings indicated that supervisors exhibiting empathy and effective communication skills engaged in collaborative efforts to determine suitable work modifications and offered ongoing support. This approach contributed to employees feeling both confident and valued in their workplace [21]. Furthermore, a longitudinal qualitative study found that employees perceive that supervisors play a key role in enabling workers to perform valuable work [22]. A study among Norwegian based on 12 nurses in Norway found that supervisors need to be attentive to employees, as this is regarded as a prerequisite for seeing, listening, showing care, and providing constructive feedback [23]. Another qualitative study found that employees highly valued leaders who demonstrated personal commitment and dedication to their work, maintained availability, provided support to the team, and adopted effective management approaches, particularly in handling complex situations [24]. These qualitative studies highlight the crucial role of supervisors in promoting employee well-being and managing occupational stress. Employees value supervisors who actively engage in work adjustments, provide ongoing support, provide genuine care, and are committed and available.
Next we have changed the following section (on page 6) to this:
However, a limitation of these studies is that most have examined strategies for managing emotional demands, primarily from the employee’s perspective. Without understanding supervisors’ perspectives, it becomes difficult to provide actionable guidelines on how to effectively manage their responsibilities in creating supportive environments. Nonetheless, few studies have addressed this gap by incorporating supervisors’ perspectives. For instance, a study by Rollins et al. (2021) involved interviews with 40 mental health clinicians and managers to explore their perceptions of organizational conditions influencing burnout and work engagement [25]. The authors identified that fostering a workplace culture centred on person-focused care, rather than solely productivity, while implementing effective strategies to navigate bureaucratic challenges and supporting employee growth and self-care, were key factors that could mitigate burnout and enhance engagement. Another study, based on semi-structured interviews with 30 Allied Health Managers, found that an empathetic leadership style—one that seeks to understand and support staff—could improve employee morale within public health organisations and reduce the risk of burnout [26]. Additionally, a study based on semi-structured interviews was conducted with 29 supervisors from 15 organisations in Australia, focusing on the strategies used by supervisors to manage employee stress. The study identified six overarching themes: four that reflect a risk assessment process model (i.e. problem identification, immediate problem execution, coping assistance, and follow-up and evaluation), and two that represent supervisory leadership behaviours that promote prevention and foster an organizational culture-supportive health [27]. Finally, a study investigated and compared stress and resilience factors in the nursing profession from the perspective of registered nurses and their supervisors. The study found that openness, trust, conflict management, and dealing with mistakes were described as requiring development in order to manage stress [28]. While these studies offer valuable insights into supervisors' perspectives on managing stress, mitigating burnout, and fostering work engagement, a gap remains in understanding how supervisors believe they can effectively address the negative impacts of emotional demands across different workplace contexts. This study seeks to address this gap.
Furthermore, this study should explicitly identify the specific workplace contexts under examination and provide a clear rationale for why these particular workplace environments merit investigation as research objects.
Our reply: We have added this in the method section on page 9: Workplace selection was carried out by identifying industries in Denmark with high emotional demands. This information was reported in the National Workplace Environment Monitoring among Employees [33], which is conducted biennially and invites approximately 50,000 employees to participate. From this data, we identified sectors with high emotional demands and, based on this, selected and invited workplaces to participate in the project.
In Section 2 (Methods), when describing the collection of non-numerical data, the survey methodology should be explicitly detailed, including the specific survey administration timeframe, data collection procedures, and a thorough justification of the questionnaire design and item selection.
Our reply: To enhance clarity, we refer to a previous study where a description of the survey administration, timeframe, data collection procedures, questionnaire design, and item selection can be found. We have added this on page 8: A description of the survey administration, timeframe, data collection procedures, questionnaire design, and item selection can be found in a previous study.
In the section 3 (Data collection), I cannot find the baseline data of the survey depending on the research of “Andersen, L.P., Andersen, D. R. Pihl-Thingvad, Jesper, Reducing the Risk of Burnout: Identifying Practical Strategies to Manage Emotional Demands - A Mixed Methods Study. 2024”.
Our reply: We are sorry that we were not clear in our communication. We had used an incorrect reference, which has now been corrected.
Additionally, while the pilot survey aims to 'identify workplaces where employees experienced high emotional demands but reported low levels of burnout,' this approach overlooks the complex, counterintuitive, and nonlinear relationship between emotional demands and burnout. The analysis should account for factors beyond internal workplace resources, including the personal meaning employees attach to emotional demands, individual emotional regulation skills, and external positive emotional connections with clients. These additional variables should be either considered or controlled for in the analytical process to provide a more comprehensive understanding of this multifaceted relationship.
Our reply: We acknowledge that the process of sampling workplaces with high demands and low burnout is not a simple process. We believe that our sampling including the core factors of experience of demands and burnout is a central demarcation for the in depth processes we wanted to explore in the current study. We agree that further factors of demarcation could improve the sampling. But provide a more comprehensive understanding of the multifaceted relationship between emotional demands and burnout, is beyond the scope of the overall project as well as the current study of burnout.
We have added this in limitation section: It may be a limitation that our sampling strategy was limited to a demarcation based on the core factors of emotional demands and burnout. It is possible that the inclusion of additional factors relevant to the complex process of burnout—such as the personal meaning employees attach to emotional demands, individual emotional regulation skills, and external positive emotional connections with clients—could have improved the sample. However, the purpose of the study was not to provide a comprehensive understanding of the multifaceted relationship between emotional demands and burnout. Rather, the study aimed to examine how supervisors support employees in managing the emotional demands of their work. Therefore, sampling based on these two central phenomena was considered key to including a relevant yet varied sample of respondents, reflecting diversity in the workforce while still ensuring the inclusion of meaningful experiences related to the process we aimed to investigate.
Comments on the Quality of English Language
There are grammatical problems in this study. For example, the sentence 'Therefore, this study aims to explore the practice though which supervisory support can help employee managing emotional demands' (p. 2, line 52) should be corrected to: 'Therefore, this study aims to explore the practices through which supervisory support can help employees manage emotional demands
Our reply: We have corrected the error and conducted a thorough review of the manuscript, during which we made several additional corrections.

Round 2
Reviewer 2 Report
Comments and Suggestions for Authors
I appreciate the thorough elaboration in the revised version. I think the manuscript now reaches the level required for publication.
Author Response
I appreciate the thorough elaboration in the revised version. I think the manuscript now reaches the level required for publication.
Our reply: Thank you very much, and once again, thank you for your constructive comments.
You mentioned that the results section and conclusion could be improved
Our reply: After a careful review, we believe that these sections meet the level required for publication, and therefore, no changes have been made. I hope you agree with us.